



# Measurement report: Rocket-borne measurements of heavy ions in the mesosphere and lower thermosphere – Detection of meteor smoke particles

Joan Stude[1,5], Heinfried Aufmhoff[1], Hans Schlager[1], Markus Rapp[1,2], Carsten Baumann[1], Frank Arnold[4], and Boris Strelnikov[3]

[1]German Aerospace Center (DLR), Institute of Atmospheric Physics, Oberpfaffenhofen, Germany
[2]Ludwig-Maximilians-Universität München (LMU), Atmospheric Physics, München, Germany
[3]Leibniz-Institute of Atmospheric Physics (IAP), Kühlungsborn, Germany
[4]Max-Planck-Institute for Nuclear Physics (MPIK), Heidelberg, Germany
[5]Royal Institute of Technology (KTH), Division of Space and Plasma Physics, Stockholm, Sweden

**Correspondence:** Joan Stude (joan.stude@gmail.com)

**Abstract.** We present data from flights of two improved ion mass spectrometers in the mesosphere and lower thermosphere region. The instruments were optimized to detect large ion masses of up to m/z 2000 and 20000 respectively, for analysis of meteor smoke particles. The flights were performed in the frame of the PMWE campaign, initiated and coordinated by IAP/Kühlungsborn, to investigate polar mesospheric winter radar echoes in Andøya/Norway in 2018 and 2021. Both flights were successful and allow to investigate the mass number and chemical composition of charged meteor smoke particles. We found a complex and divers composition of positively and negatively charged molecules and particles. While at altitudes below 85 km we observed negatively charged particles of up to several thousands of atomic mass units, above this altitude we found possible building blocks of these large particles that form right after their ablation from the parent meteorite material. While in the first flight we detected no positively charged molecules and ion clusters above m/z 100, we measured positive and negative ions with masses up to around m/z 400 in the second flight. Due to the very large mass range of m/z 20000 used in the second flight and the subsequent lower mass resolution, unambiguous mass identification is not possible. Comparing our findings to proposed meteor smoke particle compounds by other authors, our observations would be consistent with Magnetite, Fayalite and Forsterite. However, other possible compounds cannot be excluded.

## 1 Introduction

Meteor smoke particles (MSPs) range from molecule to nanometer sized particles that form due to ablation and re-condensing of meteoric matter upon entering the atmosphere. The high velocity of the impact causes the parent material to heat up such that partial or complete ablation takes place at around 90 km altitude, in the mesosphere/lower-thermosphere (MLT) region. The actual composition of MSPs is subject to a number of studies (Antonsen et al., 2019; Hervig et al., 2017; Plane et al., 2014; Robertson et al., 2014) and is still an open subject (Plane et al., 2023) which require challenging measurements in the MLT region. The DLR has developed an improved an ion mass spectrometer, jointly with the MPI for nuclear physics



and LMU Munich, which is based on an original design of MPI with numerous deployments during rocket flights in the 1970s and 1980ies (e.g., Schulte and Arnold (1992)). For the present purpose, two sounding rockets were equipped with this improved ion mass spectrometer, optimized for the detection of large ion clusters. The rockets were launched in the frame of two PMWE-sounding rocket campaigns led by IAP Kühlungsborn in April 2018 and October 2021, with a total of 4 rockets.

Here we present the results from those two rocket flights for both, positive and negative ions and discuss our measurements with regard to the detection of charged meteor smoke particles. Both flights took place in Andøya/Norway and an overview of the campaign and payloads is given in Strelnikov et al. (2021) and Staszak et al. (2021). In a previous paper (Stude et al., 2021) we introduced in detail the instrument used during the first flight in 2018. In the present paper we will describe the technical improvements implemented for the rocket flight in 2021 (see Table 1). In addition, since we concentrated in the first paper the

data presentation to selected mass spectra, we will expand the discussion to the entire data set of the rocket ascend of both flights in 2018 and 2021.

**Table 1.** ROMARA flights within campaign "PMWE"

| Flight | PMWE1F | PMWE2F |
|---|---|---|
| Date | 13 Apr 2018 | 1 Oct 2021 |
| Launch time (UT) | 09:44 | 10:03 |
| Apogee [km] | 121.4 | 126.7 |
| Mass range [m/z] | $\approx 2000$ | $\approx 20000$ |
| Mass resolution (Xe) | 17.4 | 5.8 |

Our instruments were able to obtain mass spectra between altitudes of 55 km to 121 and 127 km, respectively, during ascent and during descent, albeit then in the wake of the payload and thus with limited results. The instruments were calibrated to differently large mass ranges in order to analyze heavy ions and possible clusters of these, proposed by others (Schulte and

Arnold, 1992; Gelinas et al., 1998; Lynch et al., 2005; Rapp et al., 2005, 2012; Strelnikov et al., 2012; Robertson et al., 2014; Havnes et al., 2015; Asmus et al., 2017).

In (Stude et al., 2021) we showed selected ion mass spectra from 55.7, 69.2 and 106.3 km altitude for positive and negative ions from the PMWE1F flight. Our major findings were that we have detected large negative ions of m/z > 2000 but no heavy positive ions other than the expected proton hydrates (charged water clusters) and the layer of iron ions.

We start this report with a description of the technical differences between the two instruments (ROMARA-1 and ROMARA-2) and their calibrations, followed by an overview of the data from both flights and a presentation of selected measurement details of interest. In the concluding section we discuss our measurements with regards to a possible detection of MSP compounds that have been proposed by: Rapp et al. (2012); Plane et al. (2014); Hervig et al. (2017).



## 2 The ROMARA instruments

The ROMARA (ROcket-borne MAss spectrometer for Research in the Atmosphere) instruments are cryogenically pumped, quadrupole mass spectrometers. These type of mass spectrometers filter ions according to their mass per charge (m/z), in between 4 electrode rods, assembling a high frequency, quadrupole electric field (Paul and Steinwedel, 1953). A subsequent channel electron multiplier then detects individual charged particles and produces small current pulses that are amplified and counted by the data processing unit. To produce a mass spectrum, the oscillator voltage at the quadrupole is gradually

increased in mass channels and the respective counts are stored. A more thorough description is given in Stude et al. (2021) for ROMARA-1 and the PMWE1F flight. ROMARA-2 (flight PMWE2F) required changes that affected the aerodynamic performance and led to further small modifications. In Fig. 1 both instruments are shown side by side to illustrate these differences.

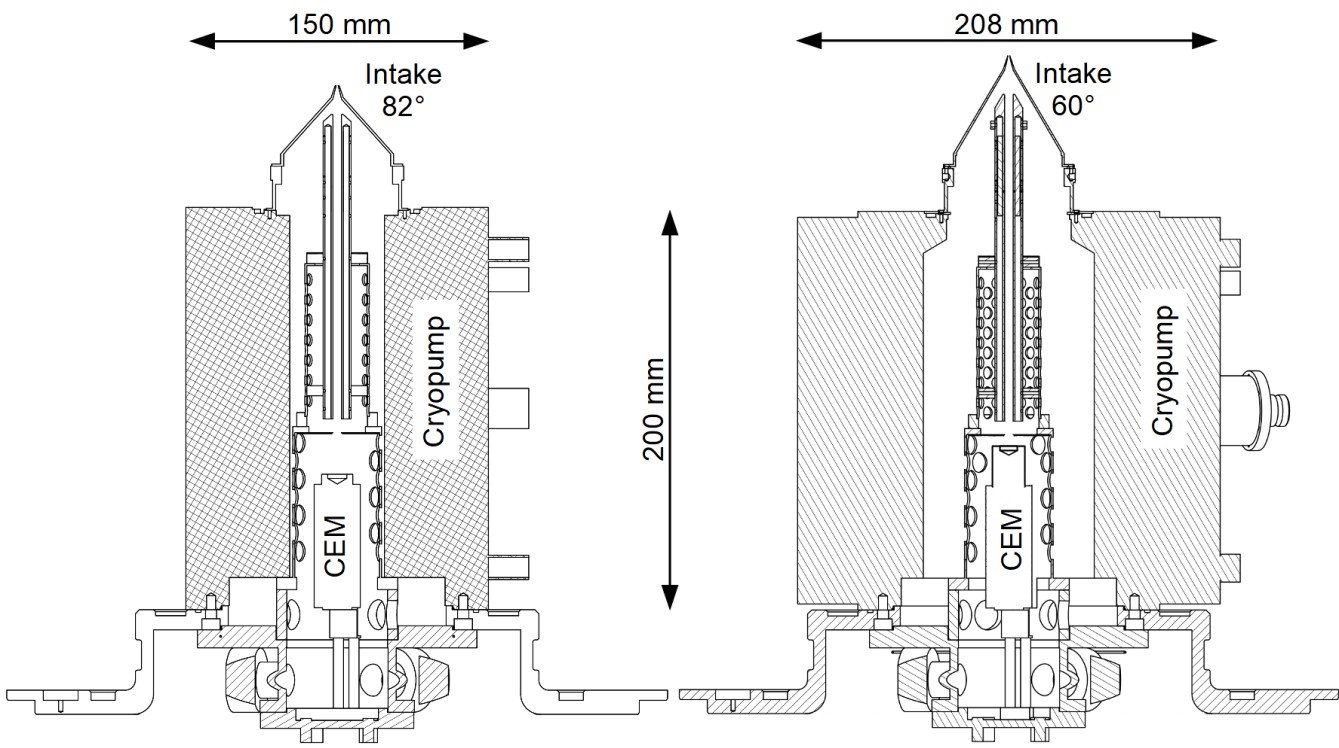

**Figure 1.** Technical drawings of the mass spectrometer ROMARA-1 (left) and ROMARA-2 (right) in their flight configuration.

ROMARA-1 used a cryopump of 150 mm diameter to accommodate enough cryogen for the required time to wait on the
launch pad for the desired atmospheric conditions. During refurbishment for ROMARA-2 we noticed a leak in the cryo system and attempts of repair did not meet the tight requirements of the vacuum system's leakage rate. We therefore used an alternative pump with 208 mm diameter from our shelves. This required adjustments in instrument geometry to avoid deterioration of the





aerodynamic performance around the intake cone. Therefor, we improved the intake cone to a more pointier design with an opening angle of $60°$ instead of $82°$ as with ROMARA-1 to regain comparable aerodynamic performance and assure an

undisturbed sampling of the atmosphere. Following the smaller opening angle of the intake cone, the quadrupole lens was lengthened by 13.3 mm to maintain same distance to the intake orifice. We further moved the channel electron multiplier up to increase detection efficiency.

## 2.1   Electronics

The electronics section of ROMARA-2 was modified as well with two major improvements: the control of the quadrupole

oscillator and the amplifier to detect the channel electron multiplier pulses. For ROMARA-1 we used A111F from AMPTEK® as charge amplifier mainly for its sufficient performance and small form factor but realized that under certain conditions the count rate might exceed the limits of the A111F and thus we upgraded to the A121 from the same company. The A121 charge amplifier allows a higher count rate of up to 12 MHz and thus has a smaller minimum dead time of 80 ns (350 ns for A111F). For ROMARA the count rate is corrected for the amplifier dead time by:

$$R_t = \frac{1}{R_m^{-1} - \tau} \tag{1}$$

with $R_t$ as true count rate, $R_m$ as measured count rate and $\tau$ as dead time. The second improvement concerned the control of the quadrupole oscillator as we wanted more flexibility for ROMARA-2. The high voltage RF oscillator amplitude $V_{RF}$ is controlled by a single analog voltage $V_c$, such that $V_{RF} = GV_c$, with $G$ being in the order of 250. As $V_{RF}$ is linear proportional to mass, so is $V_c$:

$$[m/z] = \frac{4eGV_c}{0.908r_0^2(2\pi f)^2} \tag{2}$$

with $e$ as elementary charge, $r_0$ the radius of the imaginary circle between the rods of a quadrupole and $f$ as the frequency of $V_{RF}$. As the mass resolution is roughly constant over the mass range, mass peaks or mass steps in the mass spectrum have an increasing width with higher mass settings. For a mass channel scanning with a constant step width, this means more and more mass channels are used on a peak or step at higher masses. This was the case for ROMARA-1, where a saw-tooth of

4096 voltage steps concluded in 4096 linear mass steps of roughly 0.5 u width. In the extreme case of m/z 2000 a hypothetical mass step at a mass resolution of 17.4 $m/\Delta m$ is about 115 u wide, spending about 230 mass channels on it while for m/z 30 only 4 mass channels are used. This wastes precious measuring time on a sounding rocket flight. To achieve the same amount of mass channels per step over the full mass range a logarithmic control is required, especially for an instrument with even lower mass resolution such as ROMARA-2. For the given mass resolution of about 6 and a mass range of about m/z 20000

a total number of 64 logarithmic mass channels would cover the whole mass range. With the given dwell time of 300 usec a single mass spectrum could thus be run in a fraction of the time, however the high voltage oscillator needs a certain time to settle the higher value and thus we decided for a conservative approach, leading to 256 mass channels for each spectrum with 8 x 296 usec dwell time, effectively cutting the acquisition time in half. Thereby we could keep the same data structure of the underlying data acquisition system and run two spectra in the given 4096 mass channels. Logarithmic mass steps thus allow

minimizing the time needed for a complete mass spectrum without loss of information if the mass resolution allows it.





**Table 2.** ROMARA mass filter settings

|  | ROMARA-1 | ROMARA-2 |
|---|---|---|
| frequency [MHz] | 1.4 | 0.5 |
| RF voltage [V] | 1750 | 1810 |
| radius $r_0$ [mm] | 2.13 | 2.13 |
| mass channels | 4096 | 256 |
| dwell time [usec] | 300 | 8 x 296 |

## 2.2 Calibration of ROMARA-2

As we observed during the ROMARA-1 flight a substantial negative ion population with masses above m/z 2000, we decided to increase the mass range for ROMARA-2 to about m/z 20000 by decreasing the oscillator frequency from 1.4 MHz to 0.5 MHz. This has the drawback of a reduced sensitivity and resolution, with initial tests in the lab showed about a factor 2 less sensitivity for krypton ions used for calibration. The mass resolution $m/\Delta m$ was reduced to about 6 for krypton and xenon ions, compared to 17.5 of ROMARA-1. Both instruments are operated in the RF-only mode for quadrupole mass spectrometers providing highest sensitivity. The result is a spectrum sometimes described as integral or high pass mode, where ideally all ions pass the filter when the oscillator voltage is zero and ions within the maximum mass range are filtered out when the oscillator voltage is maximum, thus letting only ions beyond the mass range of the instrument pass to the detector. The mass scan, increases the oscillator voltage gradually to the maximum and thus produces mass spectra with negative steps in the count rate until all ions are filtered out or the maximum voltage of the oscillator is reached. In practice this behavior is not strictly an integral or cumulative spectrum but rather large mass windows that are moved through the spectrum and for a reduced oscillator frequency this behavior is unfortunately amplified. At low mass settings heavy ions do not respond well to the oscillator voltage and thus are not guided well through the quadrupole filter. As the oscillator voltage increases, the ions are guided more efficiently as they respond better to the now sufficiently large fields and form a peak before they get filtered out (see Fig.2 and Fig.3).

In Fig.2 we show the response of ROMARA-2 for xenon and PFTBA ions, together with the NIST standard and a comparison to equivalent ROMARA-1 data. It is visible how some xenon ions pass at lower masses but form the aforementioned peak around m/z 80 before being filtered out at their respective mass per charge. The effect is similar on ROMARA-1 but to a much lesser extend. The xenon response for ROMARA-2 has a mass resolution $m/\Delta m$ at 50% FWHM of 5.8, while for PFTBA the mass resolution is between 4 and 8, depending on measuring mode and step size. In Table 3 the fits to the major steps of ROMARA-1 and ROMARA-2 calibration are given for comparison.

For higher masses, we simulated the general behavior of how ions generate a peak in the RF-only mode of ROMARA-2 using SIMION® and one can see how these "mass windows" move through the mass range. Ions were simulated with an angle of attack of 5° and a cone distribution inside the intake orifice. The cone angle is determined as the vector from payload speed



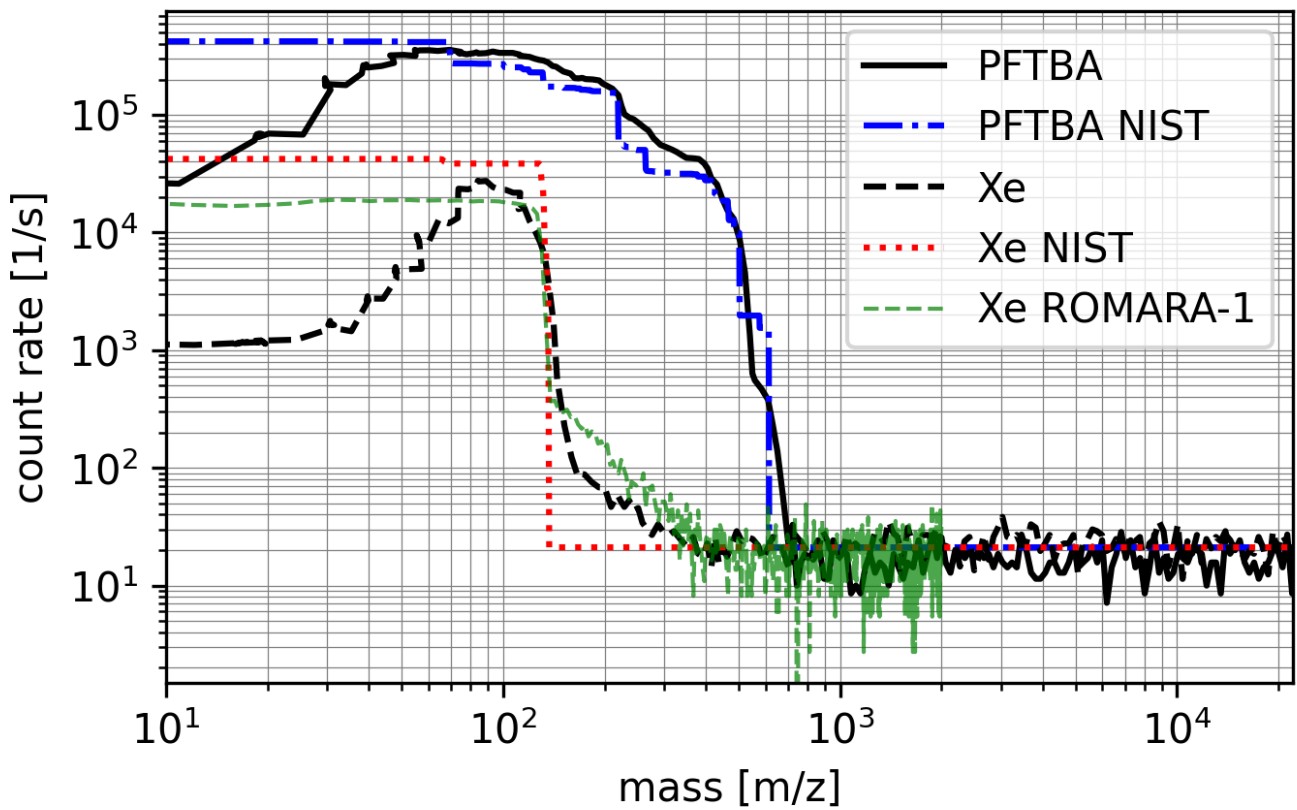

**Figure 2.** Mass spectra of ROMARA-2, for Xenon and PFTBA, the corresponding spectra from the National Institute of Standards and Technology (NIST) and a ROMARA-1 calibration for comparison

**Table 3.** Major steps of ROMARA calibrations

| NIST [u] | ROMARA-1 [m/z] | ROMARA-2 [m/z] |
|---|---|---|
| 131.3 | 130.9 ±3.8 | 123.7 ±17.4 |
| 219 | 218.2 ±3.4 | 220.2 ±18.4 |
| 264 | 257.6 ±13.8 | 259.1 ±54.9 |

(1000 m/s) and thermal speed, i.e. heavy ions have a smaller cone angle. Each falling edge means ions are filtered out, while rising edges mean the mass filter becomes transparent for the respective ions and it is thus possible that edges might disappear as they cancel each other as shown in Fig. 3.

A further finding of the ROMARA-1 flight was a not well understood saturating effect of the count rate, possibly connected to very heavy ions, resulting in a dramatic dead time increase of the channel electron multiplier. To mitigate a possible saturation






**Figure 3.** SIMION® simulation of ROMARA-2 mass response for heavy ions

we decided to include a so called mode B in the sequence of mass scans with a lower sensitivity by decreasing the bias potential of the quadrupole rods. ROMARA-1 had a bias of 20.5 and 50 Volt at the quadrupole lens and rods, which we call mode A and was used in ROMARA-2 for keeping a comparable mode, that is used on both instruments. In mode B the bias potentials were reduced to 5 and 12.5 Volt.

To prove the concept of mode B and the general sensitivity of ROMARA-2 to heavy ions we tested the mass spectrometer in the laboratory with an electrospray ionization (ESI) source before integration of the instrument into the cryopump and the rocket payload. The ESI source was adapted to our laboratory setup from an "LCQ classic" ion trap (Thermo Finnigan). In this setup only the quadrupole/detector assembly and the electronics of ROMARA-2 could be used. Figure 4 shows mass spectra of a tuning liquid (Agilent Part Number: G1969-85000) of mode A and B for negative ions. The ESI-L mass spectrum data

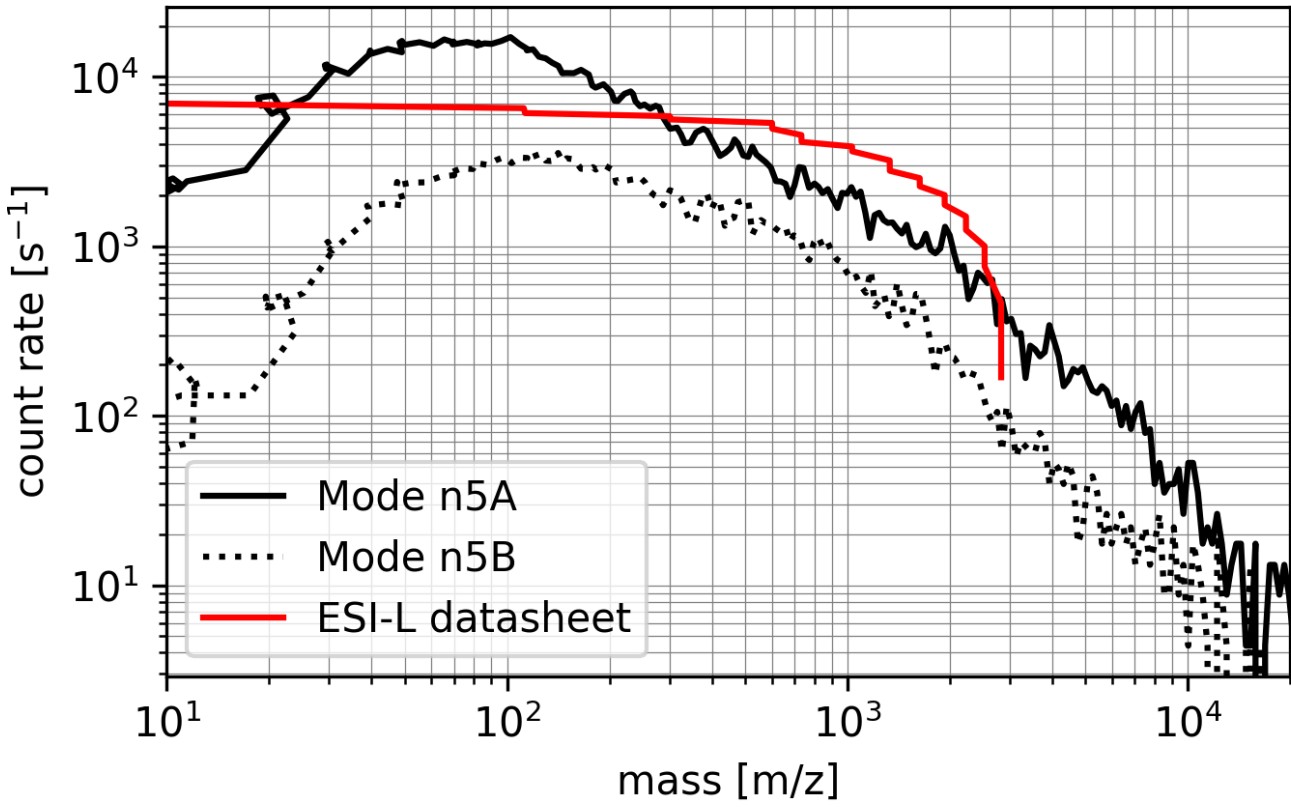

**Figure 4.** Negative ion mass spectra of the ROMARA-2 quadrupole with an electrospray ionization source (Thermo Finnigan), using ESI-L Low Concentration Tuning Mix from Agilent.

from Agilent given in its respective datasheet (Fig.4) is limited to m/z 3200 and does thus not fully cover our mass range. For a confirmation of the ROMARA-2 mass spectra, i.e. the heavy tail above m/z 3000, a laboratory mass spectrometer with sufficient mass range would be required to interface in the same way to our setup which we did not have. The maximum major ion masses correlate quite well with the mode B spectrum where the count rate declines at about m/z 3000 similar to the datasheet, although distinctive identification is not possible. Thus the results are only of qualitative nature but prove that

ROMARA-2 is sensitive to ions of several thousand masses per charge and that mode B provides some form of attenuation even if it is not constant over the whole mass range.

### 2.2.1 Measuring modes and sequence

ROMARA-1 measured positive and negative ions in alternating order with the intake cone at payload potential and a spectrum time of 1.2 s. This seemed to have provided just enough altitude resolution for the major layers in the atmosphere during the



corresponding flight. However the unbiased intake cone was a considerable risk as e.g. a negative payload potential prevents negative ions from entering the intake cone if their relative energy is below the payload potential.

For ROMARA-2 we implemented 8 different measuring modes: a positive and negative (p,n) mode, an intake cone potential of 0 (payload potential) or ±5 V (0,5) and Mode A and B (A,B). Thus the notation, e.g. "p5B" means positive ions, at -5 V intake cone potential in mode B. This notation will be used throughout the paper. As the payload charges up during flight a

bias voltage on the intake cone helps ions to overcome the payload potential. Although 8 measuring mode seems to severely decrease the altitude resolution we considered it necessary, as it might help to improve the understanding of ROMARA-1 data. As an illustration, we show the measuring sequence at around 87 km altitude of ROMARA-2 in Fig. 5. The different modes are lined up as follows: n0A, n5A, p0B, p5B, n0B, n5B, p0A, p5A. The example also shows the effect of the payload potential as the payload apparently charges up negatively and thus prevents negative ions of being detected during "n0A" and "n0B" mode.

With a spectrum time of around 600 usec and a payload speed of 1000 m/s (73 km) an altitude resolution of around 600 m is achieved repeating every 4.8 km.

## 3   Measurements

### 3.1   Rocket launches

ROMARA-1 was launched on 13th April 09:44:00 UTC into the tail of a polar mesospheric winter echo tracked on sight by the

local radar (MAARSY) between 78 and 80 km altitude. While passing the natural echo, the payload of PMWE1F created an additional artificial echo as reported in Latteck et al. (2019). The same artificial echo was reported for PMWE1D for the 18th of April 2018. The launch of ROMARA-2 on PMWE2F took place on 1st of October 2021 at 10:03:00 UTC into a relatively large and prolonged echo between 65 and 70 km, this time no artificial echoes were detected. The second payload PMWE2D was launched 3 minutes later. The launches and measurements of all instruments on both payloads were flawless and produced

expected data. However due to a parachute failure on PMWE2D, only the PMWE2F payload could be recovered from the Norwegian sea.

### 3.2   Positive ions ROMARA-1

The measurements of ROMARA-1 have been partly described in Stude et al. (2021) for 3 selected cases at 55.7, 69.2 and 106.3 km and are here covered as whole data set for the ascent (Fig. 6). The positive ion measurements show a somewhat

expected result of light ions (< m/z 100) with 3 key features: proton hydrates (charged water clusters) up to 82 km with orders of 3 and 4 water ligands [55u/73u], an ionized iron layer [56u] and $NO^+$[30u] or $O_2^+$[32u] above that. Minor steps in the spectra between 40 and 50 can be attributed to $NO^+(H_2O)$[48u] for the altitudes of proton hydrates and $MgO(H)^+$ [40/41u] at around 90 km. Some counts at our threshold limit indicate ions up to around m/z 100 but in general no heavier positive ions could be detected. Figure 6 shows these color coded, RF-only mass spectra between 55 and 115 km altitude during ascent

of ROMARA-1. The upper panel shows the full mass range up to m/z 2000 with omitted negative measurements, while the





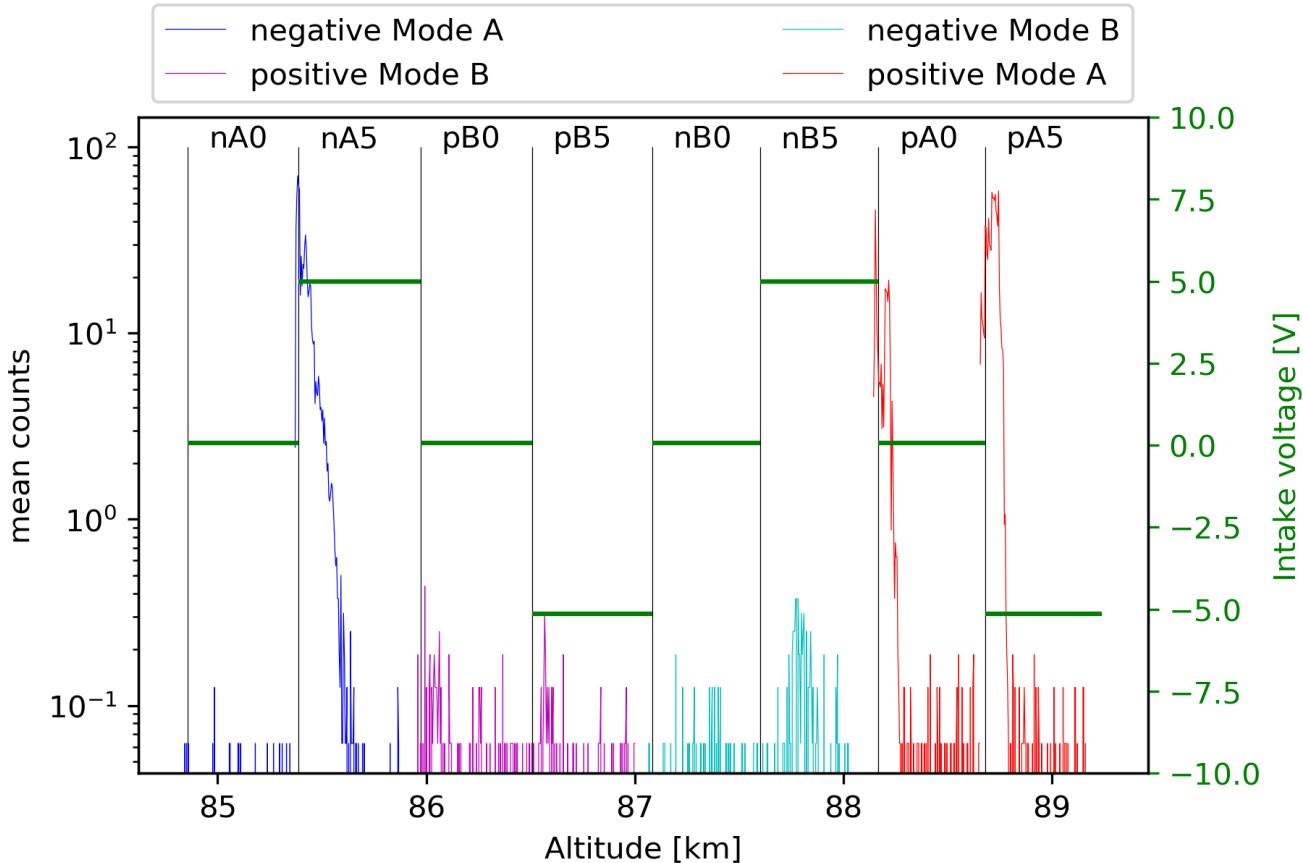

**Figure 5.** Sequence of measuring modes for ROMARA-2 around 87 km altitude. Negative payload charging prevents ions to enter the instrument in the "nA0" mode.

lower panel is restricted to m/z 110 and the negative measurement slots are filled with the positive measurements to improve visibility.

### 3.3 Positive ions ROMARA-2

The positive ion measurements of ROMARA-2 did show expected light ions of below m/z 100 and in contrary to ROMARA-1,

some unexpected signals above m/z 100, notably around 97 km, with masses between m/z 200 and 400. In the upper panel of Fig.7 we plot all positive measurements at their respective altitude at the full mass range with the negative measurements blanked out. In the lower panel we restrict the mass range to m/z 400 and fill the negative measurement slots with the closest positive data to provide a better visibility, similar to Fig.6. Further measurements of mode A at 0 V cone bias are marked in the plot as horizontal lines to indicate the sequence of the four different measurement modes (pA0, pA5, pB0, pB5) for each

charge state.



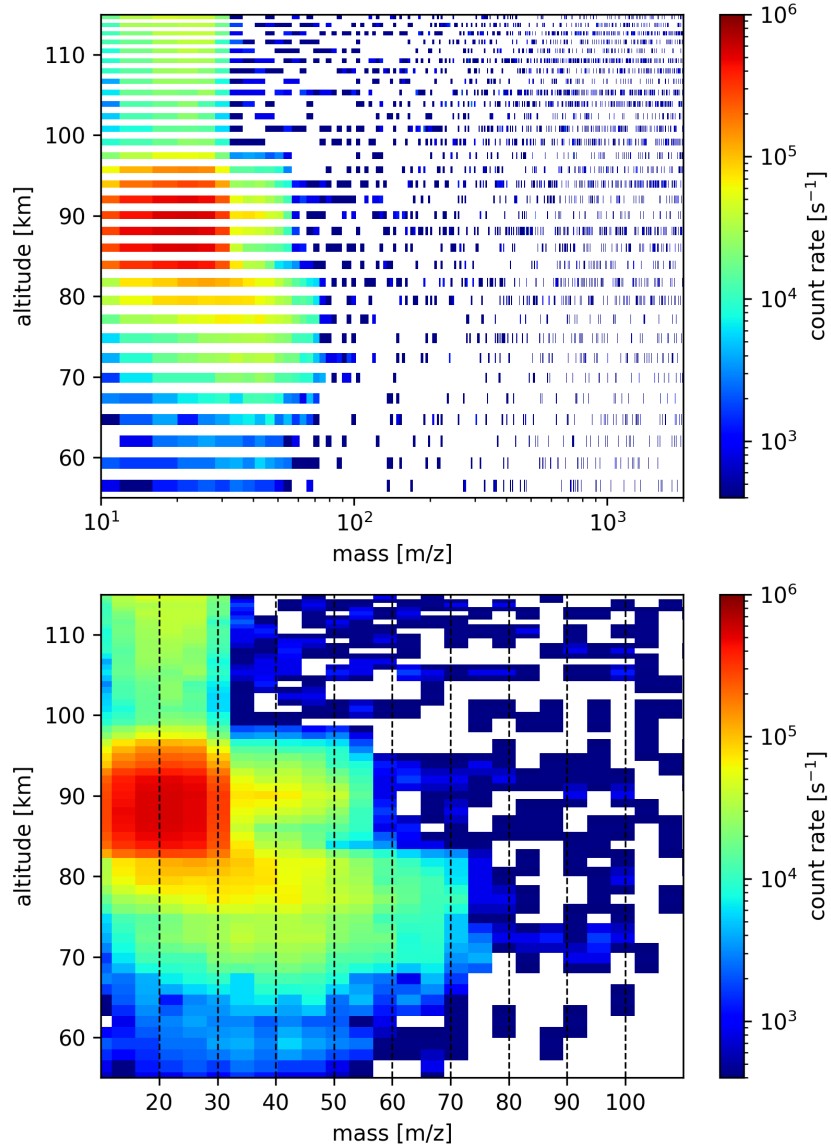

**Figure 6.** Positive ion spectra (8ch. mean) during ascent of PMWE1F (ROMARA-1). Upper panel with full mass range and logarithmic mass scale, lower panel with linear mass scale, reduced mass range and the negative measurement slots are filled with positive ion data for better visibility.

The expected strong $NO^+/O_2^+$ peak was less dominant compared to ROMARA-1 but is still visible despite the much larger mass range and the resulting difficulties in calibration at very low masses. A similar pattern as with ROMARA-1 exists in the general mass distribution. Below about 88 km the ions are heavier as proton hydrates are present. The spectrum then becomes slightly lighter for the dominating $Fe^+$[56u] and above about 100 km the step of $NO^+$[30u] or $O_2^+$[32u] dominates. However



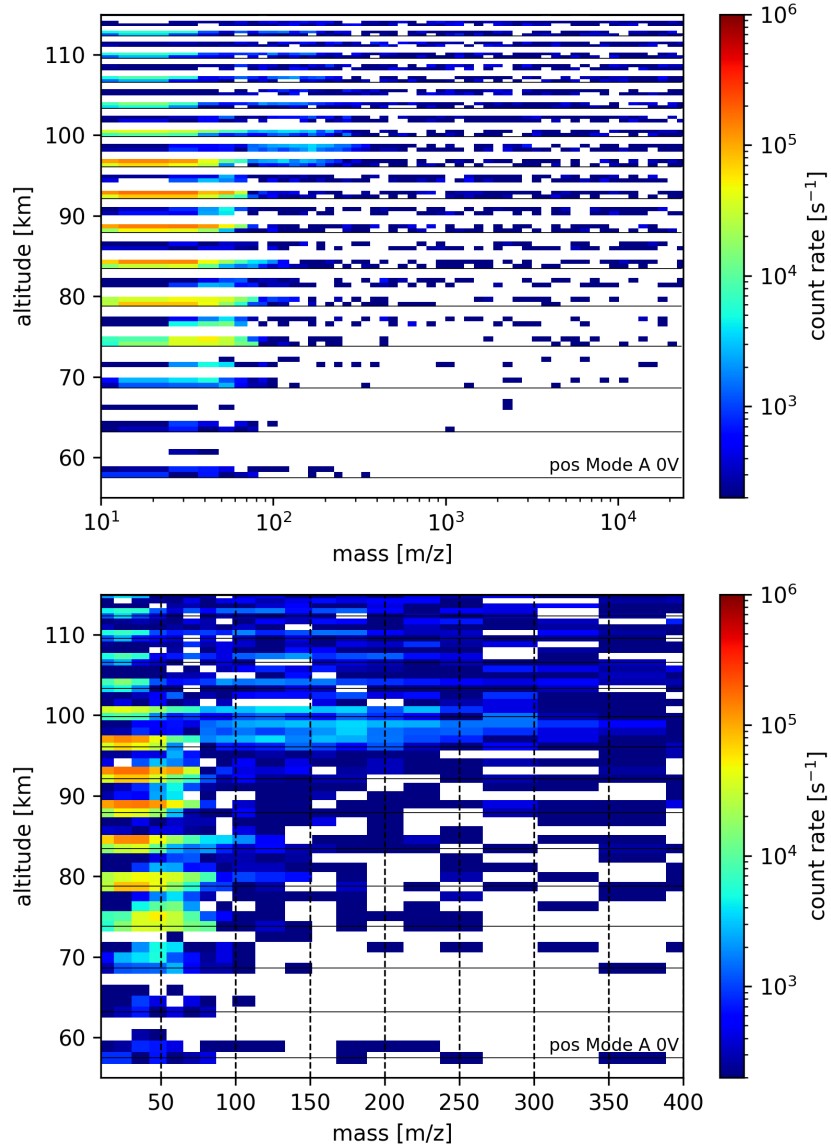

**Figure 7.** Positive ion spectra (32ch. mean) during ascent of PMWE2F (ROMARA-2). Upper panel with full mass range and logarithmic mass scale, lower panel with linear mass scale, reduced mass range and filled negative slots for better visibility. The horizontal lines mark the pA0 measurement to indicate the measurement sequence (pA0, pA5, pB0, pB5).

for ROMARA-2 the proton hydrates seem to be much heavier up to about m/z 150 at 80 km, indicating orders of 8 or 9 water ligands as reported by Björn and Arnold (1981). The striking feature, albeit with low count rates, are the heavy positive ions around 97 km just above the iron peak with masses between m/z 200 and 400. As proton hydrates at those altitudes do not form (Reid, 1977) we assume that we have detected positive MSPs.



## 3.4 Negative ions ROMARA-1

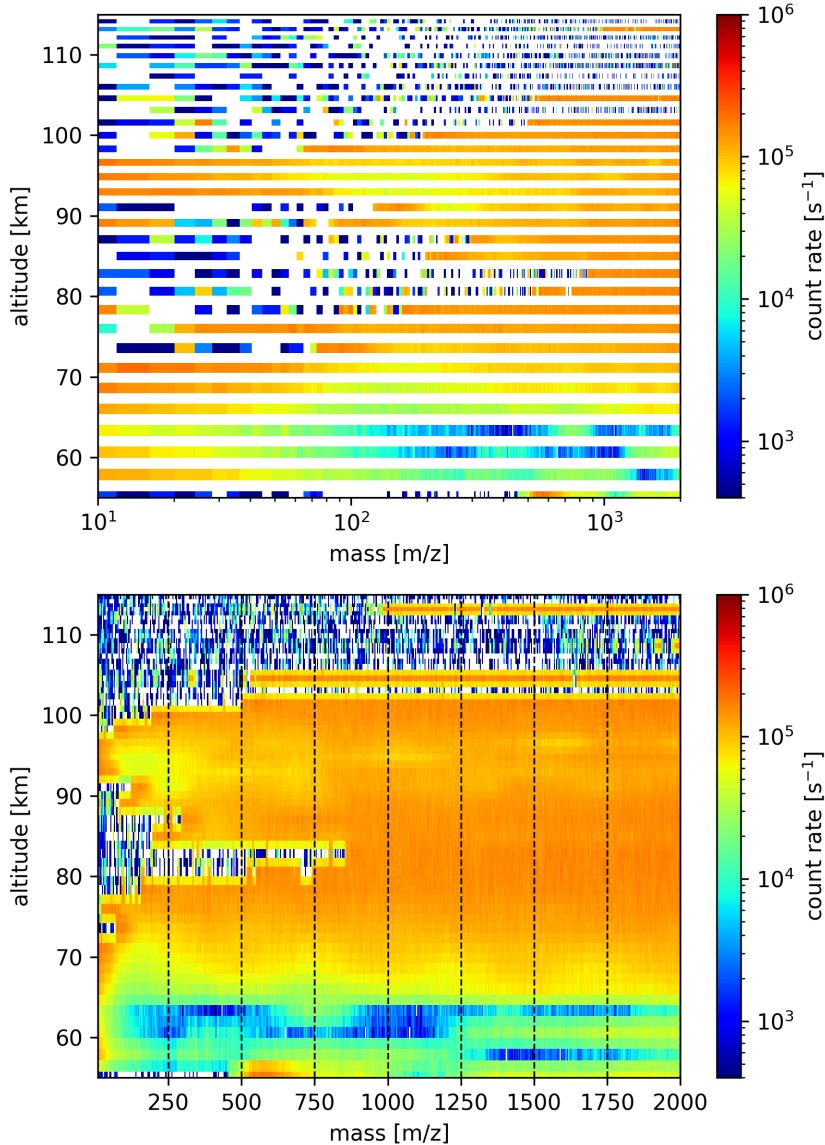

**Figure 8.** Negative ion spectra, during ascent of PMWE1F (ROMARA-1): Top panel with logarithmic mass scale, lower panel with linear mass scale and the positive measurement slots are filled with negative ion data for better visibility. Both plots have the full mass range.

The measurements of negative ions in Fig.8 show a completely different picture compared to the positive ions as our instrument was too sensitive for the prevailing atmospheric conditions. First, the moment of the cap ejection is clearly visible as heavy negative ions immediately enter into the instrument in the first spectrum of Fig.8. For the next 20 km of ascent the





signal of heavy ions is generally rising, modulated by the payload spin and then starts to saturate from about 75 km, up to about 100 km. The payload spin period is 0.27 s corresponding to a mass range of m/z 460 in the plots. The signal then disappears

and appears with the last occurrence at 114 km. Lighter ions (m/z < 150) are visible at around 70 and 95 km but due to our limited mass resolution can not be resolved or are masked as the negative channels are in general more noisy because of a higher dark count rate compared to the positive spectra. A similar analysis as with the positive mass spectra could therefore not be done. For the available spectra no distinctive steps are produced, rather a slope, which indicates that more different ions types are involved as in the positive case. Another feature is the absence of counts in the beginning of a spectrum between 75

and 90 km to varying masses, forming kind of a zero signal in the plot of Fig.8, that is peaking at m/z 850 and 85 km. This behavior could be caused by a paralyzed detector, overwhelmed with particles of masses below the reappearance of counts, e.g. below m/z 850 at 85 km. Thus we do not have continuous data for negative ions below m/z 150. Furthermore, the heavy negative ions exceeding our mass range, seem to saturate the detector at a rather low count rate for the system and thus only give a very rough maximum at about the same altitude where the detector is paralyzed the most. The low count rate could be

caused by an increased dead time as reported by Vanhaecke et al. (1998) and Gemer et al. (2020).

The exact behavior of an individual ion detector to high numbers of different incident particles can be a very complex as other parameters such as the pulse height distribution of the detector, aging or the threshold and dead time of the amplifier or electronics can play a role.

### 3.5 Negative ions ROMARA-2

The negative ion measurements of ROMARA-2 show a large variety of ion masses and most importantly confirm the existence of ions heavier then m/z 2000 as proposed before. Saturating effects as with ROMARA-1 are not existent, rather the opposite which results in very low count rates of the lower sensitivity mode B. The measurements can be broadly divided in below and above 85 km. Below 85 km the mass spectra include in general heavy ions and show little to no ions below m/z 100 which is unexpected. The majority of detected ions is between m/z 200 and 1000, at about 80 km up to m/z 5000 albeit with rather low

count rates. Above 85 km the heavy ions disappear and lighter ions are detected even in the lowest mass channels. Further, the unbiased mode with the intake cone at payload potential, does not show counts anymore, either because of the lower masses and their lower energy or because of an increased payload potential, more effectively shielding the negative ions. Above 100 km the count rates further decline due to the aerodynamic shielding effect.

### 4 Discussion

The goal of the two flights was to detect and possibly identify the chemical nature of meteor smoke particles. The data of the ROMARA-1 flight did not show a significant signal for positive MSPs but suggested a large population of heavy negative particles of > m/z 2000, albeit maybe being too sensitive and thus resulting in difficulties to interpret the negative spectra. ROMARA-2 showed, with high confidence, MSPs of both charge states. In order to identify possible meteor smoke compounds in our measurements we summarized the compounds proposed by Plane et al. (2014) and Hervig et al. (2017) in Table 4 and





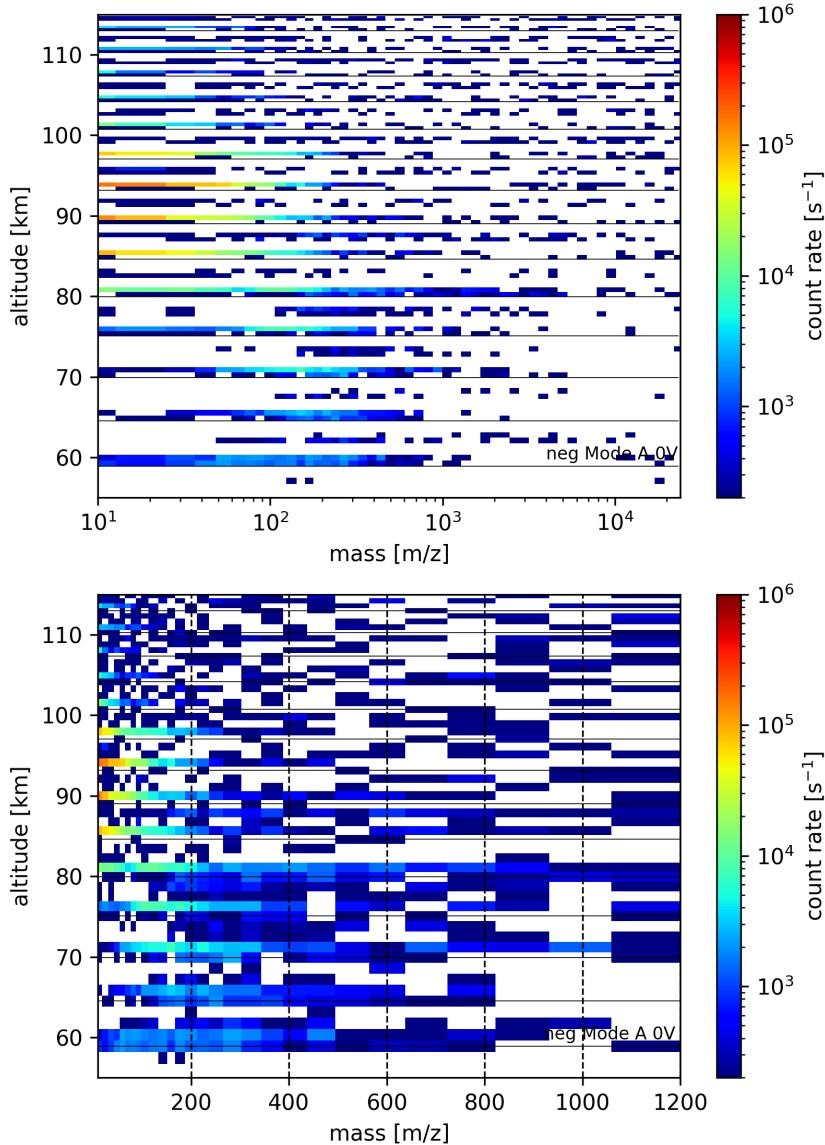

**Figure 9.** Negative ion spectra, during ascent of PMWE2F (ROMARA-2): Top panel with logarithmic mass scale and full mass range and 32ch. binning. The horizontal lines mark the nA0 measurement to indicate the measurement sequence (nA0, nA5, nB0, nB5). Positive ion measurements are omitted. The lower panel has a linear mass scale and the positive measurement slots are filled with negative ion data for better visibility.

show their atomic mass up to multiples of 3 (trimer). One can see in Table 4, that e.g. $MgO$ has such a small mass (40 u) that it's multiples would fit almost any step in the higher mass ranges with the given mass resolution of ROMARA and thus makes unambiguous identification unfeasible.





**Table 4.** Compounds as possible building blocks of MSPs by Plane et al. (2014) and Hervig et al. (2017)

| compound | mass [u] monomer | mass [u] dimer | mass [u] trimer | etc. |
|---|---|---|---|---|
| MgO[H] | 40(41) | 80(82) | 120(123) | ... |
| Mg(OH)$_2$ | 58 | 116 | 174 | ... |
| FeO[H] | 72(73) | 144(146) | 216(219) | ... |
| Fe(OH)$_2$ | 90 | 180 | 270 | ... |
| MgSiO$_3$ | 100 | 200 | 300 | ... |
| FeSiO$_3$ | 132 | 264 | 396 | ... |
| Mg$_2$SiO$_4$ | 140 | 280 | 420 | ... |
| Fe$_2$O$_3$ | 160 | 320 | 480 | ... |
| FeMgSiO$_4$ | 172 | 344 | 516 | ... |
| Fe$_2$SiO$_4$ | 204 | 408 | 612 | ... |
| Fe$_3$O$_4$ | 232 | 464 | 696 | ... |

For the discussion on the possible chemical composition of MSPs we focus on the most interesting altitudes from the individual ion spectra overview plots by averaging the mass spectra over selected altitudes. In the case of positive ions of both flights these are altitudes above the proton hydrates regime around 85 km which can be seen in the respective data. The negative ion data of the ROMARA-2 flight are examined in the same altitude range but we also analyze lower altitudes including some interesting mass spectra.

### 4.1 Discussion positive ions ROMARA-1

The most interesting altitudes for MSPs in the ROMARA-1 data are the altitudes between the iron peak at 89.6, 91.5 and 93.4 km (Fig.6). The mean spectrum from these 3 altitudes is plotted in Fig.10 up to m/z 120. One can see the aforementioned $NO^+/O_2^+$ and $Fe^+$ peak (Reid, 1977; Shuman et al., 2015; Plane et al., 2015; Kopp et al., 1984). In between these peaks, potential signatures of $NO^+H_2O$ are visible. Very few counts between m/z 70 and 100, could indicate $NO^+CO_2$[74u], FeO[H][72u/73u] or Fe(OH)$_2$[90u] as modeled by Reid (1977) but no traces of heavier ions. The picture retrieved by the instrument for positive ions thus represents a rather expected ion population, with no heavy positive ions or MSPs above our sensitivity level.

### 4.2 Discussion positive ions ROMARA-2

Despite ROMARA-2 was able to measure down to m/z 20-30 the data below about m/z 150 suffers from a larger mass number uncertainty due to the large mass range setting. The results for masses below about m/z 150 are therefore slightly shifted towards higher values as we did not aim for those masses but still wanted to see if ions are present, e.g. $NO^+$ and $Fe^+$.





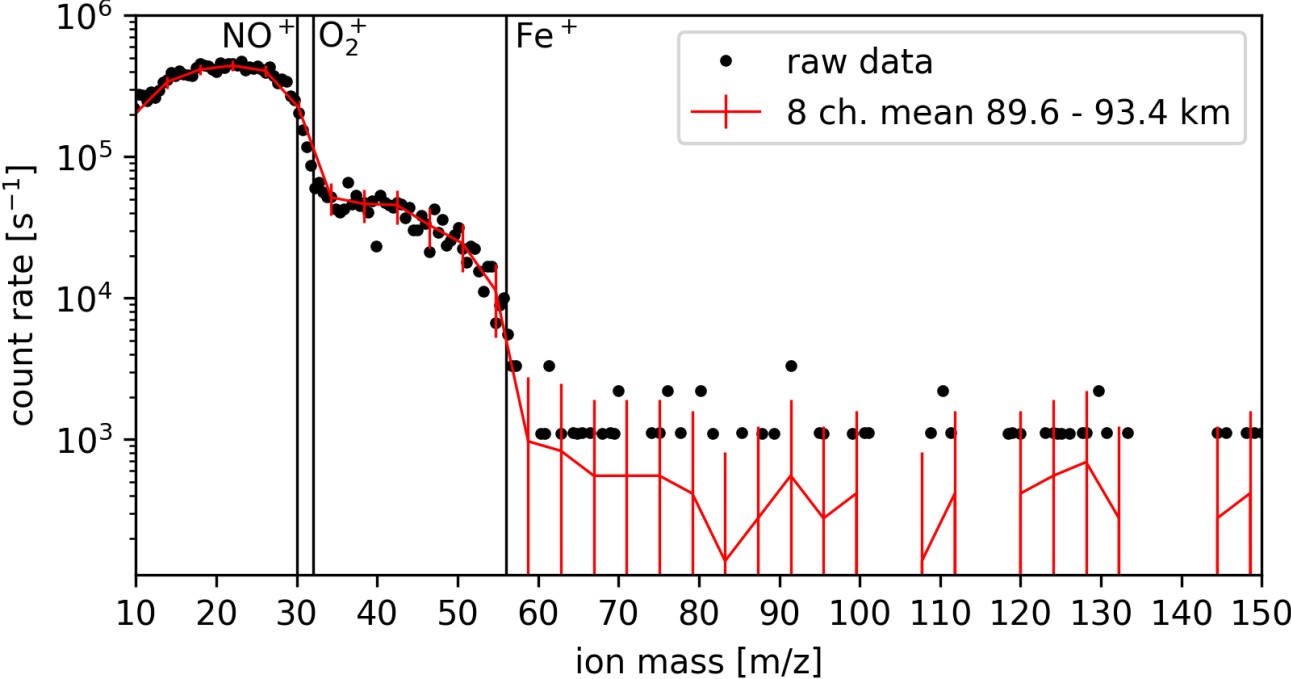

**Figure 10.** Mass spectrum for positive ions up to m/z 120 as mean between 89.6 to 93.4 km altitude during ascent of flight PMWE1F (ROMARA-1).

In Fig.7 a very interesting feature between 96.3 and 100.4 km can be seen that are likely MSPs. We plot the mean of all 6 positive mass spectra between 96.3 - 100.4 km altitude up to m/z 1000 in Fig.11. Besides the raw 8 channel data, we plot a binned set of 32 mass channels which roughly correlates to the mass resolution and provides more smooth data and is the basis of a differential spectrum. The differential spectrum is derived by subtracting subsequent channels $\Delta c = ch_n - ch_{n+1}$ and in case of a positive slope the data is omitted. The bins with the highest count rates above m/z 150 are centered around m/z 188,

237 and 302.

From Table 4 we looked at possible matches to the most probable bins we measured and show these matches in Table 5. For the mass bin m/z 188 we found $(Fe(OH)_2)_2$[180u] or $(MgSiO_3)_2$[200u], where the iron hydroxide seems to fit best as the count rate drops clearly before m/z 200. In general one might argue that compounds of lower indexes < 3, are probably more likely to exist and thus e.g. in the case of m/z 237, $Fe_3O_4$ might be a stronger candidate than $(MgO[H])_6$. But whether

$(Fe_2O_3)_2$ is more likely as $(MgSiO_3)_3$, we cannot say, except that in the same spectra plenty of iron ions are available and thus the corresponding compounds might prevail.



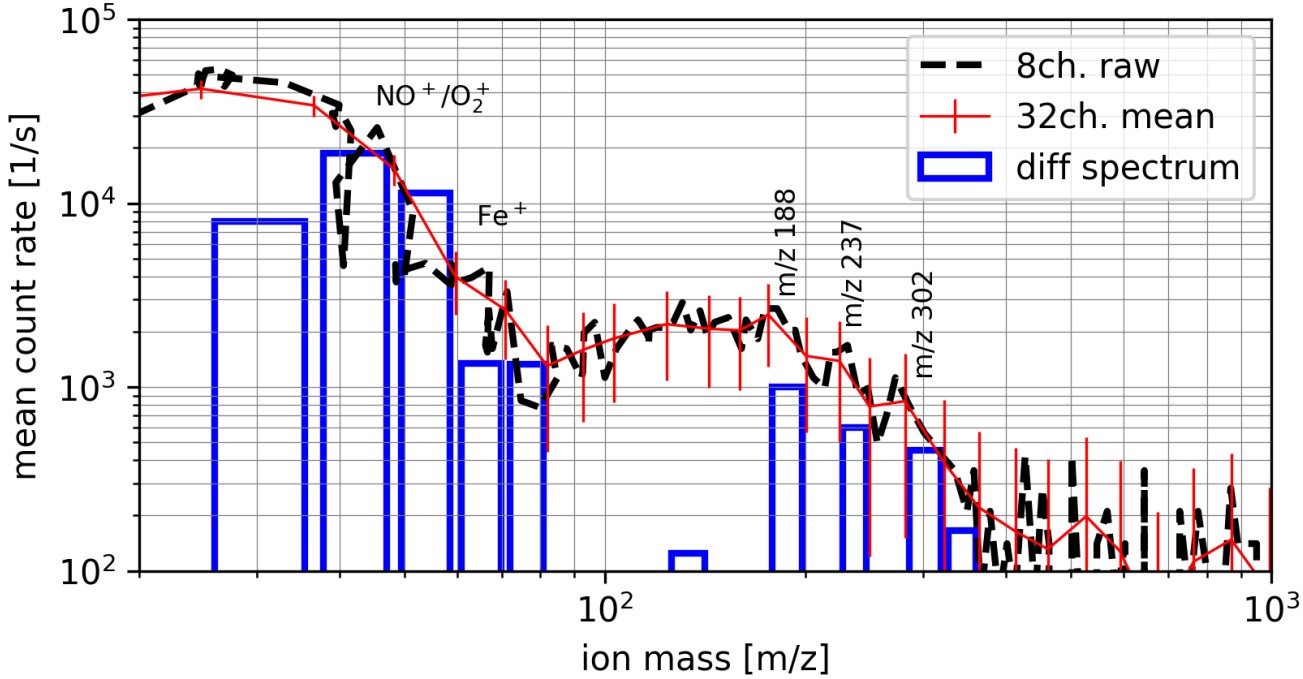

**Figure 11.** Mean mass spectra of all positive ion measurements of PMWE2F (ROMARA-2) between 96.3 - 100.4 km up to m/z 1000. Note the slightly drifted peaks for $NO^+$ and $Fe^+$. The differential bins above m/z 150 are marked with their center mass per charge.

**Table 5.** Mass bins of major steps in positive mass spectra between 96.3 and 100.4 km

| mass bin | MSP compound |
|---|---|
| 188 | $(Fe(OH)_2)_2$, $(MgSiO_3)_2$ |
| 237 | $Fe_3O_4$, $(Mg(OH)_2)_4$, $(MgO[H])_6$ |
| 302 | $(MgO[H])_7$, $(Mg(OH)_2)_5$, $(FeO[H])_4$ |
| | $(MgSiO_3)_3$, $(Fe_2O_3)_2$ |

### 4.3 Discussion negative ions ROMARA-2

As mentioned before the negative ion measurements show basically two different regions, above and below 85 km. To begin with we compare the negative ion measurements with the positive ion layer around 97 km, by taking the average of counts between 89.7 and 97.7 km in the same way as for the positive ion case. At these altitudes only the biased mode A (n5A) produced significant count rates of similar masses as the positive ions. The mean negative ion spectrum from 89.7, 93.9 and 97.7 km altitude is plotted in Fig.12, showing much more peaks compared to the positive ion spectra, which makes



identifications more difficult. For a spectrum of low mass resolution with many different steps, the spectrum turns into a slope and individual steps overlap.

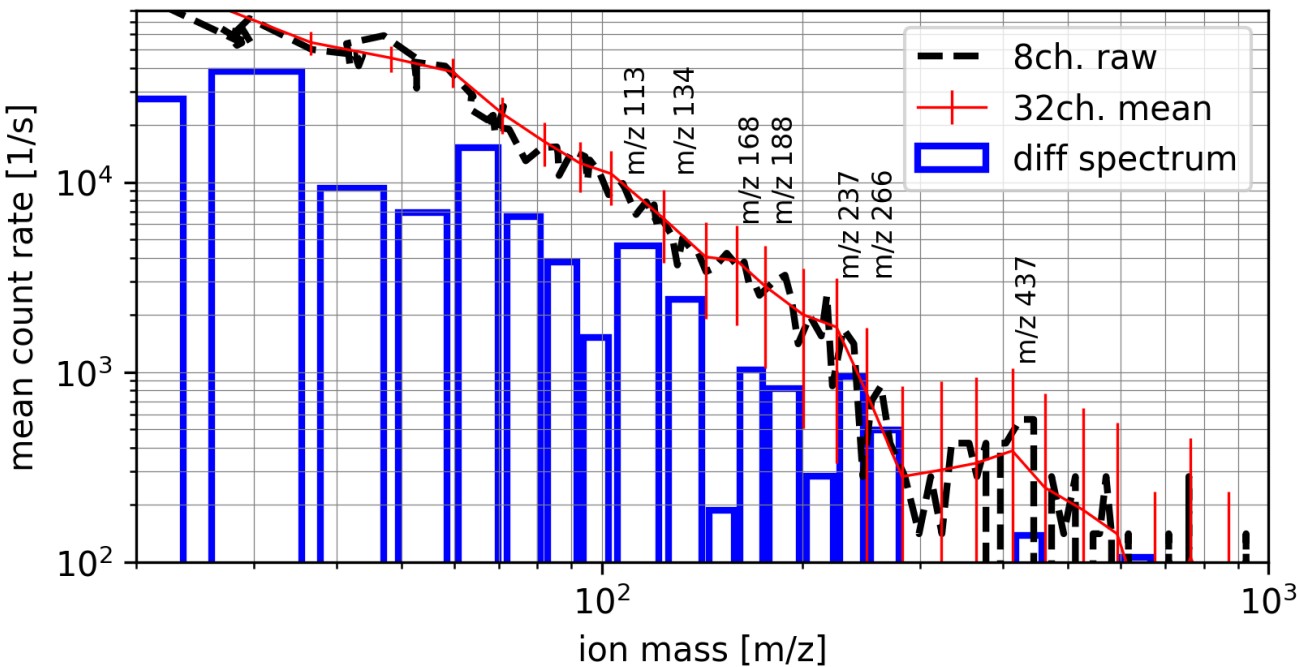

**Figure 12.** Mean mass spectra of negative ion measurements of mode n5A at 89.7, 93.9 and 97.7 km up to m/z 1000 from PMWE2F (ROMARA-2). The bins of interest are marked with their center mass per charge.

Nevertheless the possible compounds are given in Table 6. Between the possible compounds of each bin we cannot distinguish and others might rule out certain compounds. An interesting observation are the compounds $Mg_2SiO_4$ and $Fe(OH)_2$, occurring more often as possible matches, with $Mg_2SiO_4$ showing the smallest index number (trimer). The opposite is the case for $FeO[H]$, which is only once a possible match, albeit having a high probability in Hervig et al. (2017). Thus we find that $Mg_2SiO_4$ (Forsterite) is the most likely negative MSP compound for our measurements.

In Fig.13 we present 3 interesting spectra: panel a) shows a pattern of multiplying masses at 80.8 km, panel b) shows the heaviest detected mass at 80.2 km and panel c) corresponds to the altitude of the radar echo at 65.6 km, as this echo was a launch condition. The pattern in panel a) shows distinctive steps at roughly m/z 225, 450, 900 and 1800, that could indicate a compound that is doubling its mass. As the time between these mass channels is about 100 ms a correlation to the payload spin of about 3.6 Hz (277 ms) is not given. Further we do not expect proton hydrates above m/z > 400, as proton hydrates seem to

have a maximum of about 20 ligands (361 u) (Björn and Arnold, 1981). If we ignore the smaller hydroxides and oxides we arrive at 2 possible compounds as plotted in panel a) with vertical lines: $Fe_3O_4$ and $Fe_2SiO_4$ as monomer, dimer and tetramer, possibly even as octomer Thus we might have detected Magnetite (iron oxide) or Fayalite cluster MSPs. This aligns well with





**Table 6.** mass bins of major steps in negative mass spectra between 89.7 and 97.7 km

| mass bin | MSP compound |
|---|---|
| 113 | $(Mg(OH)_2)_2$, $(MgO[H])_3$ |
| 134 | $FeSiO_3$, $Mg_2SiO_4$ |
| 166 | $Fe_2O_3$, $(MgO[H])_4$, $FeMgSiO_4$, $(Mg(OH)_2)_3$ |
| 188 | $(Fe(OH)_2)_2$, $(MgSiO_3)_2$ |
| 237 | $Fe_3O_4$, $(Mg(OH)_2)_4$, $(MgO[H])_6$ |
| 266 | $(FeSiO_3)_2$, $(Fe(OH)_2)_3$, $(MgSiO_4)_2$ |
| 437 | $(MgSiO_4)_3$, $(Fe(OH)_2)_5$, $(FeO[H])_6$, $(MgO[H])_{11}$ |

the measurements and results of Hervig et al. (2017): "The most likely MSP compositions are magnetite ($Fe_3O_4$), wüstite (FeO), and iron-rich olivine (Fayalite, $Fe_2SiO_4$)".

Panel b) shows the heaviest recorded signals reaching up to m/z 5500. As the count rate is low, we refrain from assigning possible compounds but it seems that the signal between about m/z 1000 and 5500 besides being noisy does not show a typical tail, thus indicating a single compound at about m/z 5000 is producing the counts. This partly confirms the measurements of ROMARA-1 and the possibility of ions with masses above m/z 2000 although it does not explain the general observation during the flight of ROMARA-1 with a layer of large negative ions, spanning from about 65 to 100 km.

In panel c) we show the spectrum at 65.5 km, where the PMWE echo was detected. In the same way as before we found significant steps at the mass bins: m/z 188, 237 and 343 albeit the compounds at this lower altitudes could be different from above 85 km Thus for the first bin: $(Fe(OH)_2)_2$ or $(MgSiO_3)_2$, for the second bin: $Fe_3O_4$, $(Mg(OH)_2)_4$ or $(MgO[H])_6$ and for the third: $(MgO[H])_8$, $(Mg(OH)_2)_6$, $(FeO[H])_5$ and $(Fe(OH)_2)_4$ or $(FeMgSiO_4)_2$. The maximum mass in this case is m/z 500 to 600. Thus the composition at this altitude does not differ significantly from other spectra, just below or above. An

interesting RF-only spectrum for negative ions is given in Schulte and Arnold (1992) using a similar instrument at 77.6 km (Kiruna, 3rd August 1982, night time) with mass steps around 160 u, 250 u and 330 u indicating similar ions as observed with ROMARA-2.

## 5    Summary

The picture we received from our two flights is divers. Concerning positive ions during the ROMARA-1 flight we measured

a somewhat expected population of proton hydrates and $NO^+/O_2^+$, although some signs of other metal ions are missing, e.g. $Mg^+$ (Plane and Whalley, 2012) or $Si^+$ (Plane et al., 2016) but no ions above ca.m/z 100. The ROMARA-2 data essentially shows the same picture but has an additional positive ion population between m/z 130 and 350 included at unexpected high altitudes between 96.3 - 100.4 km. This unexpected layer shows a number of possible compounds, indicating a divers chemical composition, most likely revolving around iron as the most abundant metal ion.

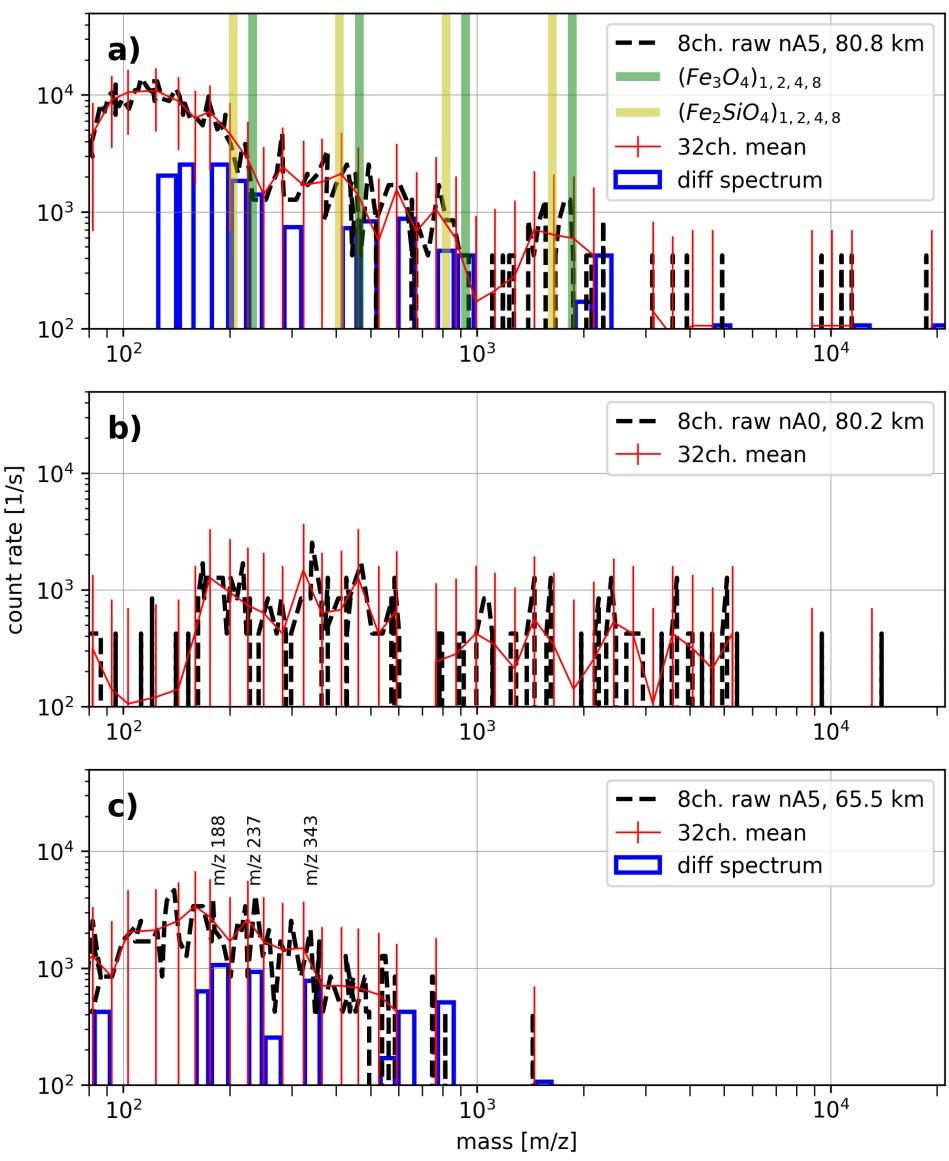

**Figure 13.** Spectra of negative ions from ROMARA-2 for the region of present PMWE (c) and of the heaviest mass signals (a,b) (PMWE2F).

Concerning the negative ion measurements we did not receive a clear picture from the ROMARA-1 data as our measurement
was saturated with ions of m/z > 2000. ROMARA-2 data could confirm that particles of such mass per charge ratio exist but
the mass range was not well suited for masses below m/z 150. The limited mass resolution of the ROMARA-2 measurements
due to the selected large mass range further restricted our analysis. However, our observations seem to be in line with the
measurements of Hervig et al. (2017) further backing up his proposals of "magnetite, wüstite, and iron-rich olivine". Rapp
et al. (2012) proposed metal hydroxides eg. MgOH and FeOH in favour of silicates which we could neither confirm or deny





for the reasons mentioned above. The overall picture from our measurements of possible compounds in MSP show a zoo of possibilities and it seems clear that there is no single MSP compound apart from a likely iron dominated chemical composition as proposed by others in the MLT-region. Unambiguously identification would require a next generation rocket-borne mass spectrometer with a mass resolution capable of distinguishing e.g. $(FeO)_4$[288u] from $(FeOH)_4$[292u].

## 6 Appenix A: Intake cone current

The intake cone current is measured with high resolution as an absolute value during the flight and is the net sum of all charge carriers interacting on it. As the cone of ROMARA-1 was on payload potential (0 V) only one curve is obtained, while ROMARA-2 had 3 different potentials (0, +5 (neg) and -5 V (pos)), as shown in Fig.14.

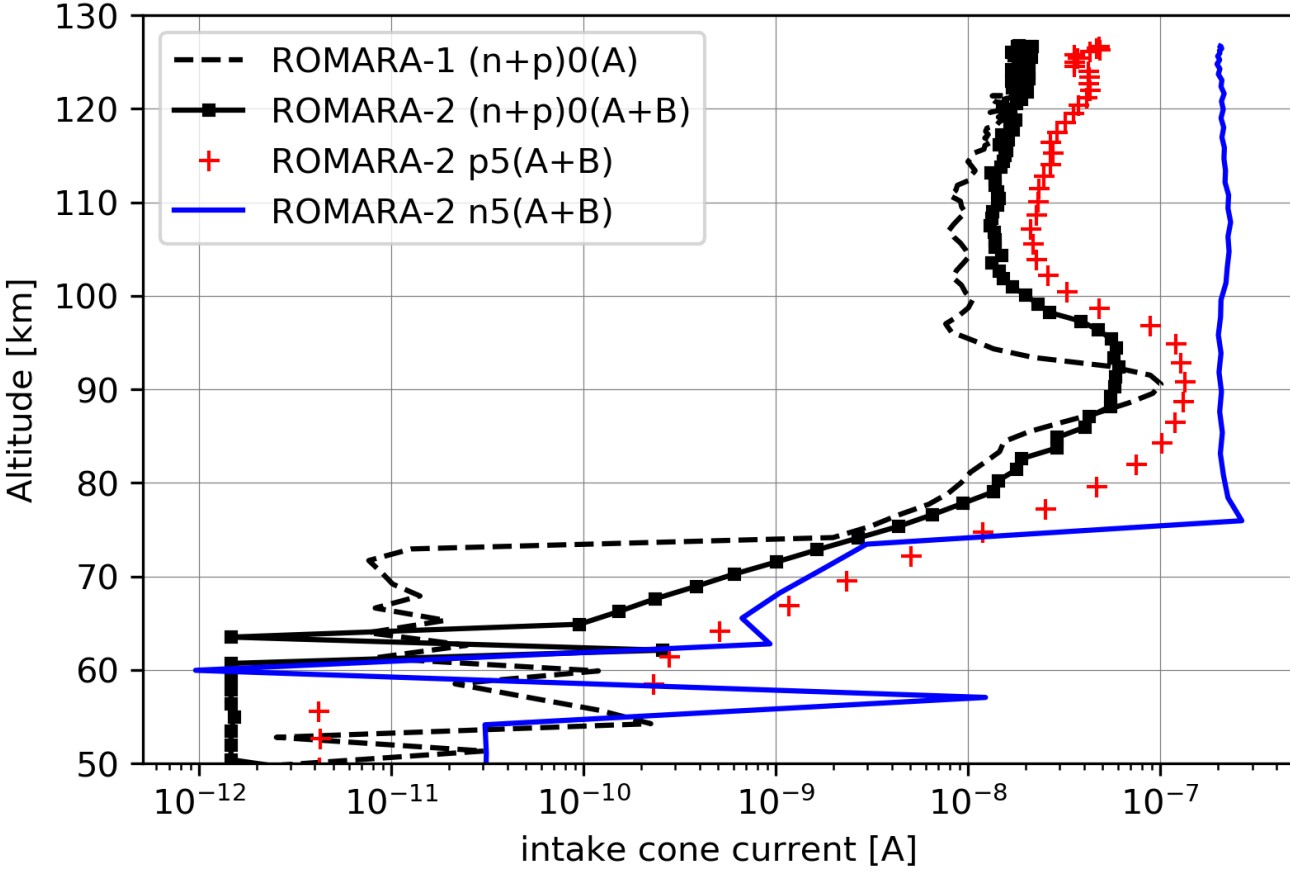

**Figure 14.** Intake cone current mean values over whole spectrum for ROMARA-1 and ROMARA-2.





# 7 Appenix B: Payload charging

An object in a plasma charges to a certain potential, such as a payload in the MLT region. Mass spectra allow to estimate the payload potential if enough ions of the right mass are present, i.e. if an ion mass is present in the 5 V mode, but absent in the 0 V mode, then the payload potential ($V_p$) must be larger than:

$$V_p > \frac{m}{2e}v^2, \tag{3}$$

with $m$ as ion mass and $v$ as payload speed. The same must be true if in both modes an ion mass is present and thus the
payload potential must be smaller than $\frac{m}{2e}v^2$. At 60 km in Fig.15 the difference in count rate between 0 and 5 V at the intake is minimal, thus a payload potential for detected ions around m/z 100 does not have an influence and gives a maximum payload potential of around -0.65 V as 100 u (lighter ions are a bit uncertain) ions would overcome the energy barrier given the payload potential. At 76 km ions of m/z 220 or 320 enter the instrument at comparable count rates for 0 V and 5 V, indicating a payload potential of around -1.1 to -1.6 V. At 85 km ions of about m/z 450 are present in the 5 V mode but not in the 0 V mode, thus
indicating a payload potential of at least -1.8 V. Thus the payload is charging up increasingly negatively to about -2 Volts as usually anticipated and previously measured by e.g. Bekkeng et al. (2013).

*Data availability.* ROMARA-1 data is available at: https://zenodo.org/doi/10.5281/zenodo.11470114 (Stude et al., 2020)

ROMARA-2 data is available at: https://zenodo.org/doi/10.5281/zenodo.11469720 (Stude et al., 2024)

*Author contributions.* JS prepared the instrument (CoPI), analysed the data and drafted the manuscript. HA prepared the instrument HS
supervision MR supervision and instrument PI FA supervision BS PMWE project PI CB ion compositions and payload charging All authors actively contributed to the discussions and to writing the final version of the paper.

*Competing interests.* The authors declare that they have no conflict of interest

*Acknowledgements.* Project PMWE was funded by the German Federal Ministry for Economic Affairs and Energy and funded by the German Space Agency (DLR) under grant 50OE1402. We further like to thank the mobile rocket base (MORABA) and Andøya Space Center (ASC)
for their intense and kind support as well as Robert Lindemann and Matthias Lang for their excellent technical work.





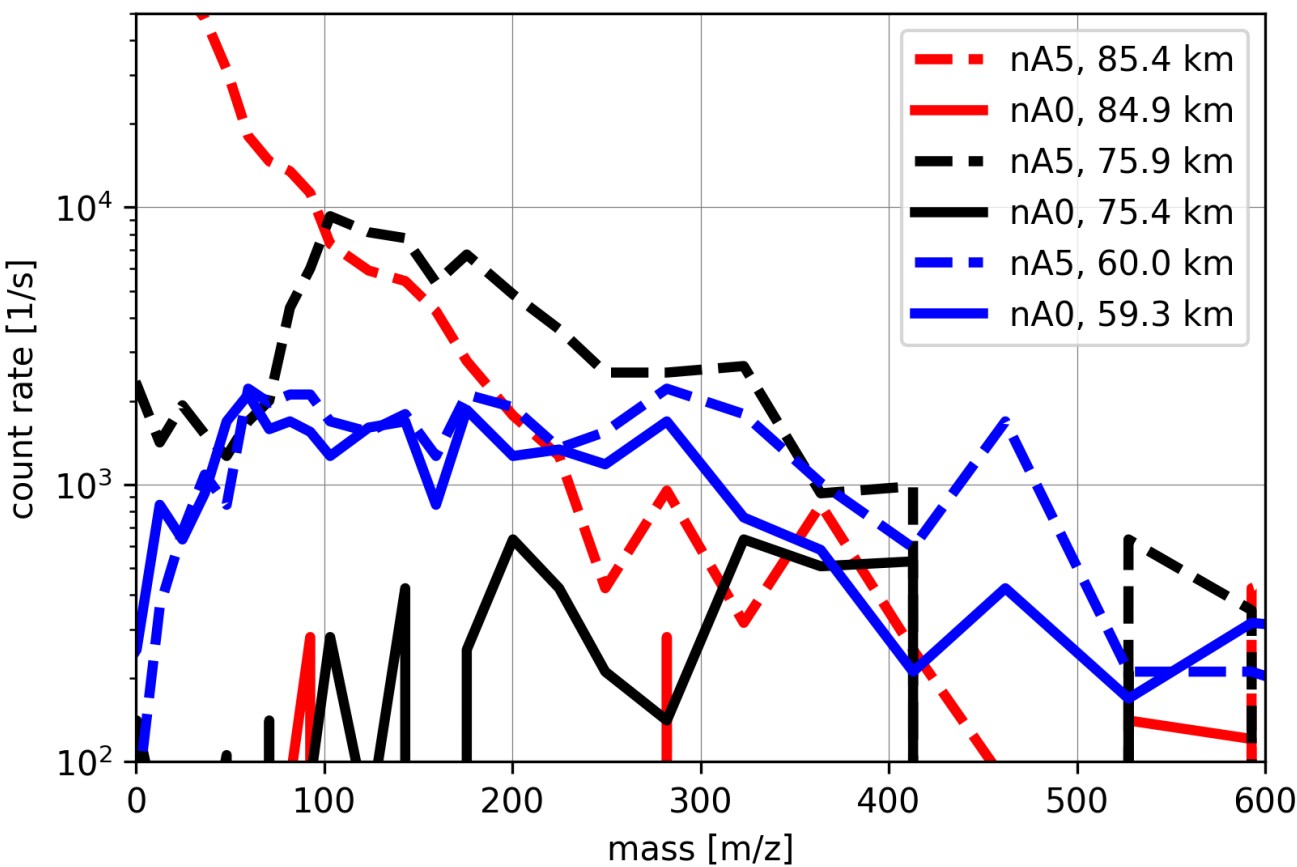

**Figure 15.** Spectra of negative ions for 0 and 5 V potentials at the intake cone, showing 3 different cases, to estimate the payload potential.

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
