# Peer review of "Measurement report: Rocket-borne measurements of large ions in the mesosphere and lower thermosphere – Detection of meteor smoke particles"

_EGUsphere, 2024_

## Author Response (AR1)

"Please upload a point-by-point response to the reviews including a list of all relevant changes made in the manuscript."

The line numbers refer to the author's track-changes file: "latexdiff_R2_20241019.pdf".

**Anonymous Referee #1**

This paper on mass spectrometric measurements of heavy ions in the meteor ablation zone adds considerably to the little that is known. On of the authors discovered these many years ago and this builds on that legacy. The paper is well written and applicable to a variety of atmospheric questions and should be published after minor revision. I have two points that will improve the paper slightly and a few grammar type corrections.

The altitude vs mass graphs include a lot of dark blue points. These seem to be ignored in some of the description. I am guessing because they are thought to be noise in the multiplier but I didn't see this description. The authors should be addressed by a short description of what is signal and what is noise.

-Added section 3.6 Background counts and Appendix C, line:242

These types of particles have been proposed to be part of a cycle that leads to meteor radio afterglows. A sentence or so mentioning this important is probably appropriate. Association between Meteor Radio Afterglows and Optical Persistent Trains**,** K.S. Obenberger, J.M. Holmes, S.G. Ard, J., Dowell,, N.S. Shuman, G.B. Taylor,  S.S. Varghese, and A.A. Viggiano *J. Geophys. Res.* **125** 10.1029/2020JA028053 (Sep 2020).

-Added reference (Obenberger et al., 2020) line:27

Page 5. Define PFTBA

-Added (Perfluorotributylamine or FC-43) line:126

Page 5, Table 2, RF Voltage maximum, not RF voltage since it is variable.

-Changed entry to "max. RF voltage [V]" line:108/Table 2

Page 9, beginning of rocket launches. I don't know what a mesospheric winter echo is, so it should be defined.

-Added "The campaigns aimed to study polar mesospheric winter echoes (PMWE) which are VHF radar echoes during the winter month at high latitudes (Latteck and Strelnikova, 2015)." line:38

Page 17, line 19 line 247, smoother not more smooth

-Changed line:280

First paragraph of summary. Divers is twice mentioned – should be diverse.

-Changed line:7 & 333

**Anonymous Referee #2**

This paper is a comprehensive presentation and analysis of recent observations of positive and negative ion mass spectra in the polar D and E region using a new cryogenic quadrupole mass filter based on a former, established design. The main objective are in situ studies of the composition of meteoric smoke particles (MSP). The authors suggest the detection of several positive and negative species, mostly containing Fe (iron), in line with models and some previous observations.

The results are based on only two flights, including some inconsistent results between the flights. More flights and further instrumental improvements should be pursued to confirm the composition altitude structure. Nevertheless, the paper is an important new contribution continuing several decades of D region ion and heavy ion measurements.

In this context I recommend to add Chesworth, E.T. and Hale, L.C. (1974), Ice particulates in the mesosphere. Geophys. Res. Lett., 1: 347-350. https://doi.org/10.1029/GL001i008p00347 as an example and review of earlier work on mesospheric heavy ions (not only icy aerosols), even though these were not all mass spectrometer measurements.

-Added reference (Chesworth and Hale, 1974) line:52

I recommend publication in its present form after some minor fixes and clarifications.

The paper is well written, however, there are some typos and other errors. I recommend using a good spell and grammar checker. A few sentences were unclear (for me or a general reader); they can be rewritten to improve the general flow and understanding.

L.3. frame —> framework -Changed line:4 & 37

L.3. PMWE introduce acronym line -Added line:39

L.4. IAP introduce acronym -Added line:5

L.5 allow —> allowed -Changed line:6

L.6. divers —> diverse -Changed line:7 & 333

where is the significant result? key points? This could be made stronger in the abstract.

Added :

- mass spectroscopic, in situ data from rocket flights line:1
- within our mass range in a region that is notoriously difficult to get mass spectroscopic data from line:8
- however, we detected positively charged particles between around m/z~180 and 350 and a number of different negatively charged particles up to m/z~5500. Line:13
- A particular interesting pattern was found at 80.8~km of a compound that seems to double its mass around m/z~225, 450, 900 and 1800. Line:16

L.20. an (duplicate) -Deleted line:32

L.20. DLR introduce acronym -Added line:32

L.20. MPI introduce acronym -Added line:33

L.21. LMU introduce acronym -Added line:34

L.22. 1980s -Changed line:35

L.25. ions, -Added line:42

L.28. the instrument in detail -Changed line:45

L.88. usec —> μsec -Changed line:102, 105, 169 and Table 2

L.100. the mass scan increases (no comma) -Changed line:118

L.100. negative steps in the count rate: unclear, what does this mean? the count rate drops?
–Changed "The mass scan thus produces mass spectra with negative steps where the count
rate drops" line:118

L.107. PFTBA introduce acronym -Added line:126

L.109.  ROMARA-1, but to … extent.  -Added comma, line:129

L.115. what is a "cone distribution"? -No changes. Particles in the 3D simulation start from a
point towards the instrument. They have a distribution of direction that is bound by a cone
with an opening angle 2 times the cone angle (not to be confused with intake cone).

L.121. capitalize Mode A, Mode B -Changed line:140, 142, 144, 148, 152, 154, 163, 197,
234, 293

L.145. mode —> modes -Changed line:164

L.171. unclear: what negative measurement slots?  I think the white gaps in Fig. 6, but can
be clearer -Changed line:196

L.181. to explain the NO+/O2+ peak; what was the total plasma density in R-1 and R-2, from
other instruments? Our measurements usually use the electron density to calibrate an ion
density. We included the electron density for R1 in Fig.14. Other data is not available and a
crude calculation from the count rate would simply be too uncertain to publish as the
sensitivity of our instrument is not well known. While we could indulge in this very interesting
topic it would defocus the paper.

L.191. too sensitive for the prevailing …   what does this mean?  Need some kind of
transition to the explanations that follow. -The negative measuring mode was too sensitive
for the present particle concentration during the flight.
–changed line:210

L.191. where is the cap ejection in Fig. 8? -Changed Figure 8 (and 6), I made a mistake with
the plots in the manuscript. The figure showed plots where I experimented with an
interpolating fill of the slots and somehow these versions entered the preprint version. This
affected Figures 6 and 8. In the corrected figure, the sudden increase of counts at about m/z
500 in the very first spectrum at the bottom of the plot, is more clear.

L.193. how can one see the payload spin at m/z 460 in Fig. 8? -Added line:215. The count rate is modulated by the payload spin with a period of about $\Delta$ m/z 460 (0,27s). This is visible between 60 and 75 km as a wavy structure. We used the mass per charge to describe the period as it directly translates from a time (mass scan), but the plot only shows mass per charge.

L.201. paralyzed = saturated? -Added reference Wuest et al. line:222

L.213. why is this unexpected? -Added line:237, in comparision to ROMARA-1

L.238. It might be helpful to reproduce a model of expected ion species from MSP here, so that the reader has a reference -No change. We refrained from including model results in the paper since we consider this to be out of scope for this measurement report. We have chosen the format of the ACP measurement report to make our measurements quickly available to all modelling groups in this field.

L.242. Despite that … -Added line:275

L.259. To begin with, (comma) -Added line 292

L.268. "high probability" according to the measurements (interpretations?) by Hervig -Changed line:301

L.273. At this time … this sentence is unclear, should be rewritten, this goes back to the explanation of spin signal in the spectra -Changed line:306, A mass scan takes a certain time, thus time correlates to mass. For R1 this is linear and for R2 it is logarithmic. However, a full mass scan in R2 needs about 600 ms and the spin rate is 277 ms.

L.294. diverse —> ambiguous? -Changed line:328

L.296. ca. —> about (circa is in English c., not ca.) -Changed line:330

L.298. diverse -Changed line:333

L.304. —> his proposed "magnetite, …" -Changed line:338

L.305. or deny —> nor reject -Changed line:340

L.308. MLT region (no hyphen) -Changed line:342

L.311. How quantitative are these total charged particle profiles? As mentioned above a comparison with an absolute electron density or positive ion density profile measured on the same payload would be helpful. -Added, We included the electron density of ROMARA-1 from Staszak 2021 into Fig. 14. Generally, the ion density is supposed to be equal to the electron density (quasi-neutral atmosphere). As the payload travels through the atmosphere at supersonic speeds the shock in front of the instrument influences the amount of particles hitting the cone drastically and thus simple relationship between cone current and ion density is difficult to establish.

L.330. add some commas or semicolons in this listing of authors -Changed line:371

**Anonymous Referee #3**

This manuscript reports the in-situ measurements of charged nanoparticles/heavy ions in the mesosphere/ lower thermosphere during two rocket flights. These are important results for research into the upper atmosphere/ionosphere because there are only few in-situ measurements of this kind. The measurements are presented appropriately. However, an evaluation of the presented measurements regarding the scientific discussion in the field is missing from the manuscript. The authors ignore theoretical work and model calculations from recent years that deal with the size and distribution of meteoric smoke particles. The connection with meteors should also be presented - albeit briefly.
-This paper focuses on the ion composition and we have chosen the format of the ACP "measurement report" to make our measurements quickly available to the science community in this field.

The reader is left with the question how the presented work connects to the current knowledge in the field.
-Our "measurement report" directly refers and builds upon significant publications in the field, which allowed us to interpret our data.

The authors seem to refer to the "meteor smoke" as discussed in the literature, but do not adequately describe what is meant by this (One small but confusing thing is that other works use the term "meteoric smoke."").
-Added references and text line:21-32

Furthermore, the terms "clusters" and "heavy ions" are used.
-Changed "heavy" to "large" ions on lines: 0, 51, 55, 123, 135, 139, 144, Fig.3, 150, 205, 212, 213, 224, 235, 237, 254, 272

A comparison of the obtained results with the particle sizes and masses used by other authors is missing. This is applicable at least for the high mass numbers and it would allow the reader to assess the results in comparison to other works.
-Added line:109, 110

The title of the manuscript seems to imply that "meteor smoke particles" are heavy ions. Water clusters are also mentioned. How are all these components connected?
-If MSPs are ionized, one could see them as a kind of heavy ion or cluster ion. Water cluster ions exists below about 85 km and are positively charged. These could be misinterpreted as MSPs in a mass spectrometer with insufficient mass resolution. Albeit at these altitudes, MSPs should be negatively charged and heavier.

Theoretical works on meteoric smoke formation often start from an initial size of 0.2 nm: based on the presented measurement, can you support this assumption?
-From Stude et al. 2021 (Fig.1) we would assume that 0.2 nm particles have a mass <100 u. Other sources give for example 0.2-0.3 nm for a water molecule. But as we measure mass and not size, molecule sizes are maybe a misleading property in the field of mass spectrometry.

How do the results link to other parameters measured during the same rocket flights?
-Added, We included the electron density of ROMARA-1 from Staszak 2021 into Fig. 14. Data from ROMARA-2 was not available.
Other measurements from the flights besides the electron density are not linked to the ion composition.

And finally, what does all this have to do with PMWE?
-Added text lines:38 - 41

The manuscript needs major revision before being published in ACM.

Minor revisions and language corrections are also recommended. Points for minor revisions were also given already by other reviewers, some are given below:

line 1: we present data from "rocket" flights -Changed line:1

line 10: m/z not defined in abstract, not defined when first used in text -Added line:3

line 14: ablation takes place over a height interval and the forming particles are carried in the atmosphere – please expand and provide references
-Added reference and text line:22 - 32

line 20/21: write out or define LMU, IAP, DLR, -Changed line:32-34

line 39: give reference for existence of water clusters -Added line:56

section 2.1: table 2 is not mentioned in the text – please also check whether all figures and tables are referred to in the text -Added line:111, 215

line 96: rephrase sentence: "The result is a spectrum…" -Changed line:114

data access: there is a link given for the data, but access does not work - possibly because of a lack of documentation
-The links to our data work. This was tested at the time of upload and again after the reviewers comments, the required information to use the data is included in the files.